# CausalConceptTS: Causal Attributions for Time Series Classification using High Fidelity Diffusion Models

## Abstract

Despite the excelling performance of machine learning models, understanding their decisions remains a long-standing goal. While commonly used attribution methods from explainable AI attempt to address this issue, they typically rely on associational rather than causal relationships. In this study, within the context of time series classification, we introduce a novel model-agnostic framework to assess the causal effect of concepts, i.e., predefined segments within a time series, on specific classification outcomes. To achieve this, we leverage state-of-the-art diffusion-based models to estimate counterfactual outcomes. Our approach compares these causal attributions with closely related associational attributions, both theoretically and empirically. We demonstrate the insights gained by our approach for a diverse set of qualitatively different time series classification tasks. Although causal and associational attributions might often share some similarities, in all cases they differ in important details, underscoring the risks associated with drawing causal conclusions from associational data alone. We believe that the proposed approach is widely applicable also in other domains to shed some light on the limits of associational attributions.

## 1 Introduction

Machine learning has achieved remarkable success across diverse fields, thanks to the development of powerful hardware and the collection of large datasets. Time series data, widely present in domains such as natural sciences, medicine, and life sciences (Wang et al., 2023; Esteva et al., 2019; Miotto et al., 2018; Shen et al., 2017; Bepler & Berger, 2021) serve as invaluable resources for modeling temporal patterns and dependencies, particularly in widely considered classification settings (Rajkomar et al., 2018; Wang et al., 2019). However, complex models such as deep learning models often sacrifice interpretability for performance, a trade-off that can be critical in downstream tasks (Somani et al., 2021; Roy et al., 2019).

**Need for explainability** A lack of insights into the model's decision making process often represents a significant hurdle when it comes to the the deployment of deep learning models in particular in safety-critical domains. This led to the emergence of the subfield of explainable artificial intelligence (XAI), see (Lundberg & Lee, 2017; Montavon et al., 2018; Covert et al., 2021) for reviews. Existing literature on XAI for time series classifiers has explored various methods (Crabbé & Van Der Schaar, 2021; Raykar et al., 2023; Zhao et al., 2023; Rojat et al., 2021; Ismail et al., 2020). However, the majority of the proposed methods rely on associations whereas ultimately one is rather interested in uncovering causal effects. Moreover, a clear understanding of the precise differences between these two kinds of attributions, both on a theoretical level as well as on an empirical level, is lacking.

**Need for causal insights** Counterfactual inference is a type of causal reasoning that involves estimating the effect of a particular intervention or treatment on an outcome by comparing it to what would have happened if a certain intervention or treatment had been applied. In medical applications, counterfactual inference has been used to estimate the effect of a treatment on a patient's health outcome (Gillies, 2018). As nicely laid out in (Goyal et al., 2019), causal attributions provide a clear advantage in the case of correlated features. The hypothetical scenario where the classifier bases its decision only on one of two correlated features cannot

be resolved with associational attributions. Therefore, associational attributions possibly fail to capture the actual model behavior.

**Main contributions** In this paper, we introduce *Causal Concept Time Series Explainer (CausalConceptTS)*, a novel model-agnostic causal attribution method specifically designed to enhance the interpretability of time series classification tasks via causal concepts, represented as predefined segments within the time series. More specifically, our main contributions can be described as follows: (1) We formalize the difference between causal and associational attributions for diverse concepts within time series data (2) We demonstrate how counterfactual outcomes, required for causal attributions, can be estimated using state-of-the-art diffusion models. (3) We conduct a comparative analysis of causal and associational attributions for a diverse set of time series classification tasks, highlighting the necessity to overcome purely associational attributions for more reliable model insights.

## 2 Related work

**Time Series classification** The taxonomy of traditional machine learning techniques and algorithms for time series classification is extensive, encompassing various approaches such as distance-based methods (Rakthanmanon & Keogh, 2013), feature-based techniques (Fulcher & Jones, 2017), interval-based models (Deng et al., 2013), shapelet-based algorithms (Hills et al., 2014), and dictionary-based methods (Schäfer, 2015). In addition to these traditional methods, numerous deep-learning techniques have been proposed for time series classification. These leverage different backbone architectures, including Convolutional Neural Networks (CNNs) (Ismail Fawaz et al., 2020), Recurrent Neural Networks (RNNs) (Karim et al., 2017), self-attention mechanisms (Rußwurm & Körner, 2020), and most recently state space models (Gu et al., 2022). This work, rely on the latter architecture, but stress that the proposed method is applicable to any classifier model, including non-deep-learning models.

**Deep generative models** The generation of synthetic time series data with deep learning has been implemented in various contexts such as conditional generation (Alcaraz & Strodthoff, 2023b), class imbalance (Hssayeni, 2022), anomaly detection (Bashar & Nayak, 2020), imputation (Tashiro et al., 2021; Alcaraz & Strodthoff, 2023a), or explainability (Goyal et al., 2019). While early backbone architectures involve VAEs and GANs, diffusion models have recently emerged as powerful generative alternative (Tashiro et al., 2021; Alcaraz & Strodthoff, 2023a). We therefore also rely on diffusion models to generate high-fidelity time-series counterfactual outcomes. Importantly, our proposed method is not limited to using diffusion models; it can be applied with various generative models and interventions, making it versatile for different scenarios.

**Counterfactuals for time series data** Counterfactual analysis in time series classification is essential for explaining model decisions because it allows us to ask "what if" questions about how changes in the input sequence affect the classification outcome. Several approaches have been explored for utilizing counterfactuals to handle time series data. Ates et al. (2021) experimented with multivariate settings for individual treatment effects, but their approach involves random sampling from appropriate training set samples, leading to discontinuous counterfactual samples. Delaney et al. (2021) proposed an instance-based framework that intervenes in samples until they belong to a different class of interest, however, the intervention areas are limited to neural network findings extracted via class activation mappings. Li et al. (2022) utilized motif discovery for identifying intervention areas, which represents a rather limited scenario due to its focus on precisely recurring patterns. Wang et al. (2021) introduced a framework for generating counterfactuals from the latent space of neural networks, capable of learning both low- and high-level concepts; however, it is only applicable to univariate time series data. To the best of our knowledge, we are the first to use high-fidelity diffusion models to estimate counterfactual time series inputs.

**Attribution methods for time series** Attribution methods are particularly valuable in time series data because they help highlight which parts of the sequence contribute most to the model's decision, where in time-sensitive applications this can improve trust based on the most relevant aspects of the data. Attribution methods for time series range across diverse downstream tasks as classification (Crabbé & Van Der Schaar, 2021), and forecasting (Raykar et al., 2023). For recent reviews we refer to see (Zhao et al., 2023) for post-hoc methods, emphasizing backpropagation, perturbation, and approximation methods and (Rojat et al., 2021) for ante-hoc methods.

As already briefly mentioned above, the existing attribution methods focus almost exclusively on associational effects as opposed to the proposed approach, which aims to infer causal effects. In this respect, the most closely related prior work is the work of Goyal et al. (2019). They use a variational autoencoder to infer counterfactuals, albeit in the context of image classification models. They mostly rely on manually defined attributes as concepts, whereas our concepts are defined in combination with specific subsets of the input.

## 3 CausalConceptTS: Causal Concept Time Series Explainer

In Figure 1, we present a schematic representation of the proposed approach. In the following paragraphs, we introduce the key concepts in detail.

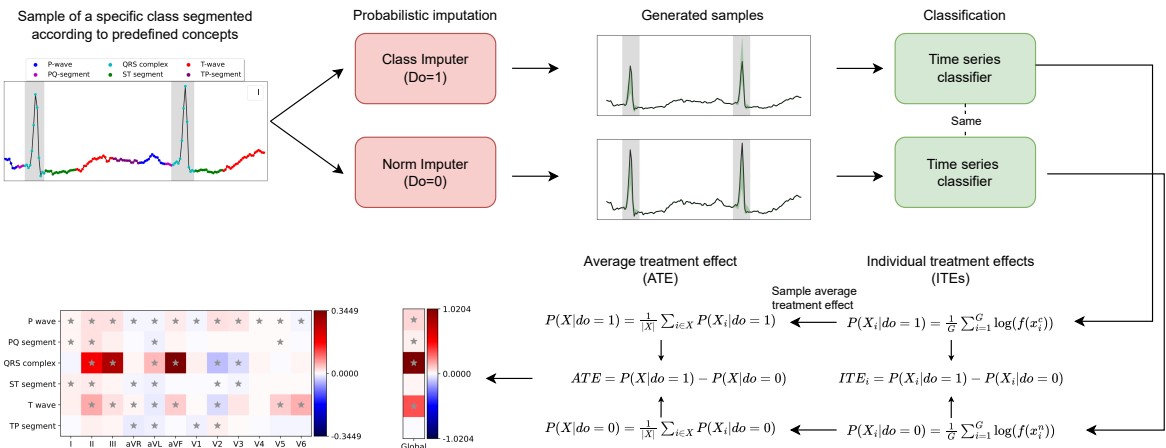

Figure 1: Schematic representation of the proposed *CausalConceptTS* approach: We start from a sample corresponding to a specific class, segmented according to predefined concept, which can either be expert-defined (such as ECG segments) or simply inferred by clustering. For a chosen concept, we impute corresponding concepts using two different imputation models, one trained on samples corresponding to the original class and one corresponding to a baseline class of choice typically associated with healthy controls, yielding two sets of imputed samples. These two sets are passed through a predefined classifier of our choice that we aim to investigate. The log difference of the corresponding mean output probabilities yields an individual treatment effect or causal attribution quantifying the causal effect of the concept in question on a specific classifier output. Sample-averaged ITEs yield corresponding average treatment effects (ATEs), which we visualize in terms of channel-agnostic as well as channel-specific causal attribution maps.

**Causal data generating process** Building on the causal attribution work in image data (Goyal et al., 2019), we adopt the causal data-generating process proposed by (Schölkopf et al., 2012). We assume for each sample (with $l$ time steps and $k$ channels hence represented as two-dimensional matrix) $X \in \mathbb{R}^{l \times k}$ that there is a class state $CS$, parameterized through several binary indicator variables, that characterizes the sample. We envision that the data generating process proceeds in two stages: In a first step, a semantic mask $M \in [1, \ldots, C]^{l \times k}$ which assigns every token in the input sequence to one of $C$ concepts, is generated conditioned on a given class state $CS$. In a second step, we denote the numerical values of the time series where the semantic mask takes the value $c \in \{1, \ldots, C\}$ as $X_M^c$, i.e., $X_M^c = \{X_{ab} | M_{ab} = c\}$. Then $X = X(X_M^1, \ldots, X_M^C)$ can be reconstructed via $X_{ab} = X_M^c[k]$ where $c = M_{ab}$ and $k = |\{(i, j) | M_{ij} = c \text{ and } (i, j) \preceq (a, b)\}|$, where $\preceq$ denotes lexicographic ordering. Furthermore, we assume that $X_M^c$ is generated through a structural causal model $h_X^c$, i.e., $X^c = h_X^c(M, CS, \epsilon_X^c)$ from the semantic mask $M$, the class state $CS$ and a noise variable $\epsilon_X^c$. We aim to use the so-defined data generating process to study a predefined (binary) classifier $f$ that maps the input sequence $X$ to an output probability $f(X)$. A visualization of the causal graph underlying our study is shown in Figure 2.

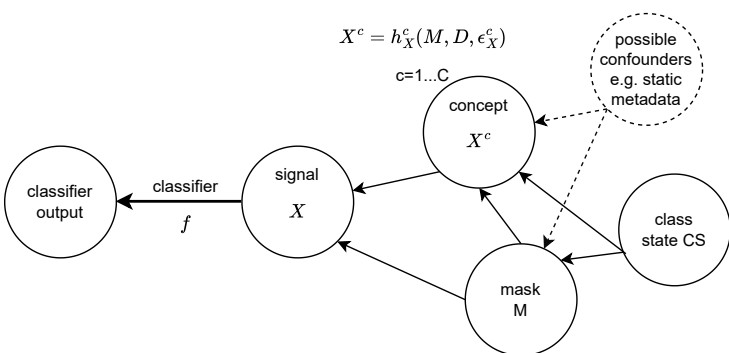

Figure 2: Causal graph underlying our approach. The data generating process is rooted in a class state $CS$, which causes a (concept) mask $M$. The class state $CS$ in combination with the mask $M$ define the specific numerical values $X_M^c$ (for concept $c$), which in combination leads to the input signal $X$. $X$ is passed through a predefined classifier $f$. We investigate the causal effect of $X_M^c$ on the classifier output by intervening on the class state $CS$. In our experiments, we neglect the causal effect of the class state $CS$, i.e., keep the concept mask $M$ unchanged. Similarly, we do not take into account possible confounders such as static metadata that could influence $X_M^c$ or $M$ which are expressed as dashed lines.

**Individual and average treatment effects** We now aim to investigate the causal effect of the class state $CS$ on the classifier $f$ by intervening on $CS$. As a crucial simplifying assumption, we assume that the underlying semantic mask $M$ remains unchanged under this intervention for a specific sample, i.e., we only intervene on the level of the generating process of the signal segments. We intervene by setting the class state to a specific value $CS^*$. As reference value we consider a baseline state $CS^0$ (in a medical context associated with healthy control samples). Then the *individual treatment effect (ITE)* for sample $X$ of concept $C \in [1, \dots, C]$ on the classifier $f$ is defined via do-operators as in (Shalit et al., 2017):

$$\begin{aligned}
\text{ITE}(X, f, c, CS^*, CS^0) = & \log_2 E_{h_X^c} f(X(X_M^{c\complement}, (X_M^c | \text{do}(CS = CS^*)))) \\
& - \log_2 E_{h_X^c} f(X(X_M^{c\complement}, (X_M^c | \text{do}(CS = CS^0)))) ,
\end{aligned} \tag{1}$$

where we use the shorthand $X_M^{c\complement}$ to denote $\{X_M^1, \dots X_M^{c-1}, X_M^{c+1}, \dots X_M^C\}$. Here, we adopted logarithmic differences instead of ordinary differences to compare output probabilities, as discussed in (Blücher et al., 2022) in the context of associational attributions. The expectation value in Eq. (1) refers to the data-generating process $h_X^c$. Below, we will use a high-fidelity generative model to sample from $h_X^c$. By averaging over samples, we obtain the *average treatment effect, i.e.,*

$$\text{ATE}(f, c, CS^*, CS^0) = E_{X \sim \mathcal{X}(CS^*)} \text{ITE}(X, f, c, CS^*, CS^0) , \tag{2}$$

where $\mathcal{X}(CS^*)$ refers to the data distribution of samples with label $CS^*$.

**Individual associational effect** Note that the individual treatment effect shows a strong structural resemblance to the PredDiff attribution method (Blücher et al., 2022), which can be considered as a special case of the Shapley value formalism where only a single coalition (the complement of the feature set $X_M^c$ under consideration) contributes. In analogy to Eq. 1, we define an *individual associational attribution (IAA) as:*

$$\text{IAA}(X, f, c, CS^*, CS^0) = \log_2 f(X) - \log_2 E_{x_M^c \simeq k_X^c} f(X(X_M^{c\complement}, x_M^c)) , \tag{3}$$

where the expectation value refers to the conditional distribution $k_X^c \equiv p(X_M^c | X_M^{c\complement})$. The IAA coincides with the PredDiff attribution for $X_M^c$.

**Relation between causal and associational attributions** We can now compare Eq. 1 and Eq. 3 to identify differences and similarities between causal and associational attributions. The first term in Eq. 1 refers to the observed outcome. We therefore expect that $\log_2 E_{h_X^c} f(X(X_M^{c\complement}, (X_M^c|\text{do}(CS = CS^*)))) \approx f(X)$ if $CS^*$ coincides with the true label of the sample $X$. The second term in Eq. 1 refers to the counterfactual outcome. The main difference between the *causal* ITE from Eq.1 and the *associational* attribution from Eq.3 boils down to the use of a class-conditional imputer (conditioned on the background state $CS^0$) in the case of the causal ITE,

$$E_{h_X^c} f(X(X_M^{c\complement}, (X_M^c|\text{do}(CS = CS^0)))) \approx \int f(X(X_M^{c\complement}, x_M^c)) p(x_M^c|X_M^{c\complement}, CS^0)\mathrm{d}x_M^c \tag{4}$$

compared to using a (class-)unconditional imputer in the case of the associational IAA,

$$E_{x_M^c \simeq k_X^c} f(X(X_M^{c\complement}, x_M^c)) = \int f(X(X_M^{c\complement}, x_M^c)) p(x_M^c|X_M^{c\complement})\mathrm{d}x_M^c \tag{5}$$

The insights from this paragraph allow us to empirically compare causal and associational attributions on the level of individual samples. The relation in Eq. 4 is only approximate as it only captures the dependence of the generative distribution on $D^0$ but neglects a dependence on other possibly confounding variables such as static patient metadata. In the supplementary material, we demonstrate for a simple example involving a single concept that associational attributions can lead to unnatural outcomes, where the attribution changes sign, while the causal attribution shows no such behavior.

**Generative model architecture** Here we elaborate on the specification of the generative model utilized for sampling from either the unconditional distribution $h_X^c$ or the conditional distribution $k_X^c$. This can be read off most explicitly from Eq. 5 and Eq. 4 respectively, where we approximate the respective right-hand side by sampling from a generative model, in an imputation task setting. For our specific implementation, we leverage the recently proposed structured state-space diffusion (SSSD) model for time series imputation (Alcaraz & Strodthoff, 2023a). This model, a diffusion model, extends the popular DiffWave architecture (Kong et al., 2021) by employing two S4 layers instead of bidirectional dilated convolutions, thereby enhancing its capability to capture long-term dependencies. Alongside a modified diffusion procedure wherein noise is applied solely to the concepts to be imputed, this approach yielded state-of-the-art results for time series imputation across various domains. To train a class-conditional diffusion model for a specific class, we simply subsample the training set to include only samples of the desired label, proceeding as in the class-unconditional case. It is important to stress that we approximate the true (but unknown) generative distribution $h_X^c$ or $k_X^c$ via sampling from an empirical imputer, which will inevitably lead to a sampling error. One can infer prediction intervals leveraging a large number of imputations from the probabilistic imputation models as demonstrated in (Alcaraz & Strodthoff, 2023a). At least for the in-distribution case, where the class-conditional imputation coincides with the sample's true class, one could even turn these into statistically valid prediction intervals using conformal prediction techniques (Angelopoulos & Bates, 2021). However, this approach will not allow to derive statistically valid prediction intervals in the counterfactual case, where the class-conditional imputation does not match the label of the original sample. The ability to generate counterfactuals from class-conditional generative models will always assess the generative models slightly outside their training distribution and hence will not enable statistical coverage guarantees.

**Generative model details** The imputation model employed within *CausalConceptTS* incorporates 36 residual layers and 256 residual and skip channels, while keeping further hyperparameters unchanged compared to (Alcaraz & Strodthoff, 2023a). We optimize the mean squared error (MSE) using the Adam optimizer, with the model undergoing 200 diffusion steps via a linear schedule. We approximate the expectation values in Eq. 4 and Eq. 5 through sampling from an appropriate generative model. The number of considered samples is an important hyperparameter. Our experiments showed convergence after around 15 samples on average due to the generative model's probabilistic nature. Consequently, we maintain generating 40 samples per real sample to ensure robustness. Training details and additional details on the computational complexity can be found in the supplementary material.

**Channel-specific attributions** When assessing channel-specific attributions, we do not condition on inputs from other channels captured at the same time as the channel to be imputed, to avoid issues with correlated channels at identical time steps, see also the discussion of interaction effects for associational attributions in (Blücher et al., 2022). Consequently, we consistently utilize an imputer trained in a blackout-missing manner. Subsequently, we substitute channels not intended for imputation with their respective values from the original dataset.

**Classifier model architecture** Building on recently successful applications in the context of physiological time series (Strodthoff et al., 2024; Wang & Strodthoff, 2023; Saab et al., 2024; Alcaraz & Strodthoff, 2024), we also leverage structured state space models (with four layers) as classifier models (Gu et al., 2022). For optimization, the Adam optimizer is utilized with a learning rate and weight decay both set to 0.001. The learning rate schedule is maintained constant throughout training. A batch size of 64 samples is used for each training iteration, spanning a total of 20 epochs. The training objective is to minimize the binary cross-entropy loss. During training, we apply a model selection strategy on the best performance (AUROC) on the validation set which usually converges before the total epochs. For the test set, we report the 95% confidence intervals obtained through bootstrapping over 1000 iterations. For additional details on the classifier model, we refer to the supplementary material.

**Concept discovery and concept validation** At first glance, the proposed approach may appear to depend heavily on predefined concepts where researchers and data owners should define concepts based on domain knowledge. However, many time series datasets lack such predefined annotations. Therefore, as a concept discovery approach, in cases where expert-annotated concepts are unavailable, we employ k-means clustering using raw time series data as input and the squared Euclidean distance as the distance measure to identify concepts. We acknowledge that this necessitates to assess the semantic meaning of the identified concepts in a second step. How this second step can be realized will be very specific for the data domain at hand. Exemplarily, we illustrate this process for an EEG use-case below.

We determine the number of clusters using the elbow method. To assess if the identified clusters are class-discriminate, we use a simple concept validation step. To this end, we conduct classification using gradient-boosted decision trees (XGBoost), using simple statistical features extracted from the cluster concepts as input. These employing six sample-wise and channel-wise concept statistics namely, minimum, maximum, mean, standard deviation, median, and number of time steps. Ideally, higher model performance indicates that these concepts effectively distinguish between classes.

**Uncertainty quantification in ATEs** The fact that we approximate the expectation values for causal/associational effects in Eq. 4 and Eq. 5 through finite samples from a corresponding imputation model allows us to infer not only point estimates of the corresponding effects from the corresponding sample means but also gives us access to the uncertainty estimate at the level of ITEs or IAAs and then correspondingly also at the level of average causal effects. Specifically, we conduct 1,000 bootstrap iterations by sampling with replacement from the test set to compute 95% ATEs prediction intervals. We claim a statistically significant causal effect if the prediction interval inferred in this way does not include the value 0.

## 4 Experiments

**Structure of this section** We conduct our experiments using a diverse range of time series classification tasks. Specifically, we present results for three tasks derived from various qualitative time series data sourced from the meteorological and the physiological domain. We present our primary experimental findings through figures, each illustrating either the associational or causal attributions. In these visualizations, we provide two attribution: on the right, we present the 'global' causal effect, encompassing the impact across all channels collectively; on the left, we delineate the channel-specific computation of the treatment effect for each concept. When considering uncertainty quantification, a star symbol indicates statistically significant causal effect in the sense of a 95% prediction interval that does not the value 0. To visualize the considered concepts, we present an exemplary plot of a time series from the dataset under consideration superimposed with corresponding concept assignments. To foster more research in this field and enhance usability for applications, we are making the source code used in our investigations available in a suitable repository (Anonymous, 2024).

**Goal of this study** In this work, our primarily focus is on the comparison of associational against causal attributions. Therefore, we only compare to associational PredDiff attributions, which are the associational analogues of the proposed causal attributions. For a comparative assessement of associational attributions, we refer the reader to the literature (Bluecher et al., 2024). Similarly, is important to stress that our model provides causal attributions for a given classifier. Under the assumption that the classifier accurately captures the true relationships in the data, we directly compare these attributions to the ground-truth causal relationships established in the literature, aiming to align our model's outputs with these known causal effects in the underlying data.

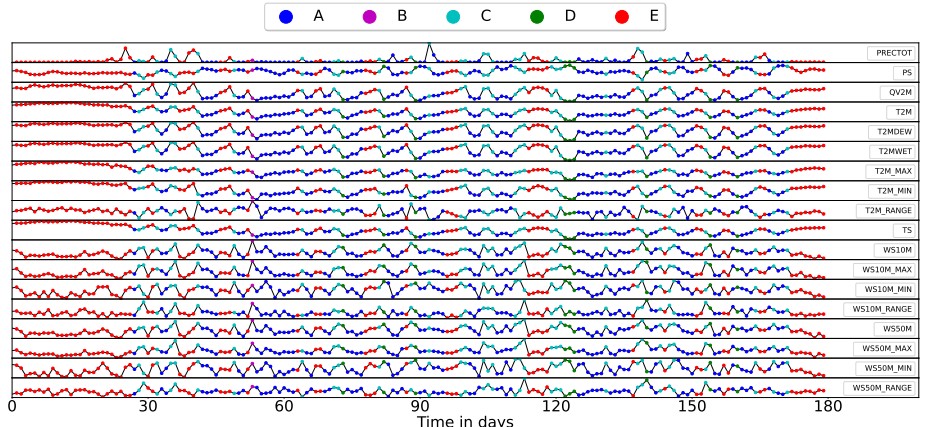

Figure 3: Schematic representation of the concepts for the drought dataset

**Drought prediction** As first task, we explore the drought dataset (Minixhofer, 2021), sourced from the U.S. Drought Monitor. This publicly available dataset involves classifying, in a binary manner, whether the upcoming week will experience drought conditions based on six months of daily sampled meteorological data. The dataset contains 18 features (Precipitation PRECOT, surface pressure PS, humidity, temperature, Dew/Frost point, wet bulb, as well as minimum and maximum temperature all at 2 meters QV2M, T2M, T2MDEW, T2MWET, T2M_MAX, T2M_MIN, T2M_RANGE. Earth skin temperature TS. Wind speed at 10 and 50 meters with their corresponding maximums, minimums, and ranges respectively WS10M, WS10M_MAX, WS10M_MIN, WS10M_RANGE, WS50M, WS50M_MAX, WS50M_MIN, and WS50M_RANGE). In the absence of expert concepts, we identify five concepts (A-E) through k-means clustering leading to an AUROC 0.7447 (95% PI 0.7406-0.7483) during concept validation. We report a classification performance for the S4 model of 0.8941 (95% PI 0.8919- 0.8962). Figure 3 visualizes concept assignments for a drought sample.

Figure 4 shows (A) associational and (B) causal attributions for the drought prediction task. Interestingly, both channel-wise attribution maps reveal a diverse range of variables with significant effects, yet they sometimes disagree on whether the effects are positive or negative. One notable observation is precipitation, which shows the highest positive effect in the causal setting but appears negative in the associational setting. Extensive research has validated the positive significant impact of precipitation on drought prediction (Cancelliere et al., 2007; Anshuka et al., 2019) which is the largest positive attribute for causal, whereas associational effect is negative across several concepts. Similarly, in concept E, a group of variables at 2 meters have been shown to have positive effects, including humidity and dew/frost point temperatures (Behrangi et al., 2015), as well as wet bulb readings, which causal attributions properly account for them while associational do not. Additionally, for concept A, factors such as the minimum, maximum, and range of wind speed at 50 meters have been shown to have a positive influence (Štěpánek et al., 2018), which again causal unlike associational attributions properly attribute to.

**ECG classification** As the second dataset, we leverage the PTB-XL dataset (Wagner et al., 2020; Goldberger et al., 2000), which is a publicly available dataset of clinical 12-lead ECG data (I, II, III, aVR, aVL, aVF, V1-V6). Although PTB-XL provides annotations in terms of diverse hierarchical levels of ECG statements in a multi-label setting, we keep the setup simply by restricting ourselves to investigation of the causal concept effects of inferior myocardial infarction (IMI) in a binary classification setting against healthy controls

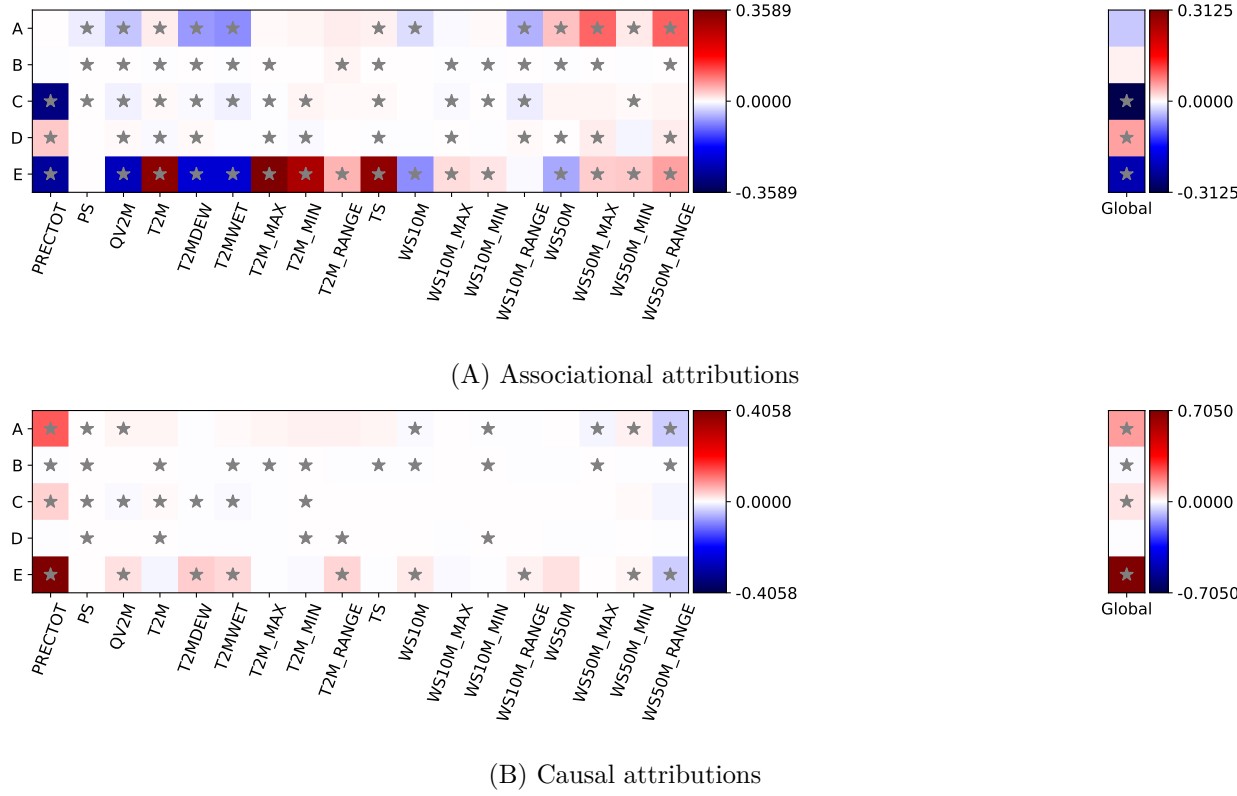

Figure 4: Illustration of the (A) associational and (B) causal attributions on the drought dataset

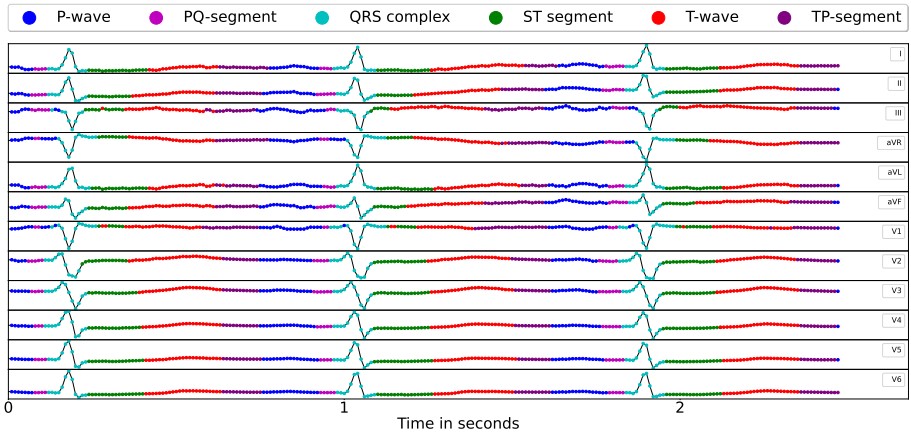

Figure 5: Schematic representation of the concepts for the PTB-XL dataset

(NORM+SR). We utilize a sample length of 248 time steps and for the predefined segmentation of the signal into channel-specific ECG segments, we leverage segmentation maps provided by (Wagner et al., 2024). Here, we consider six concepts: P-wave, PQ-segment, QRS complex, ST-segment, T-wave, and TP-segment, which reach an AUROC score of 0.9287 (95% PI 0.913-0.9435) during concept validation. The classifier reaches an AUROC classification performance of 0.9722 (95% PI 0.9621-0.9797). Figure 5 shows a visual representation of these concepts for a myocardial infarction sample.

Figure 6 presents both associational and causal attributions for the ECG classification task. The literature extensively covers this task, allowing us to draw conclusions on the channel level. Both attribution maps

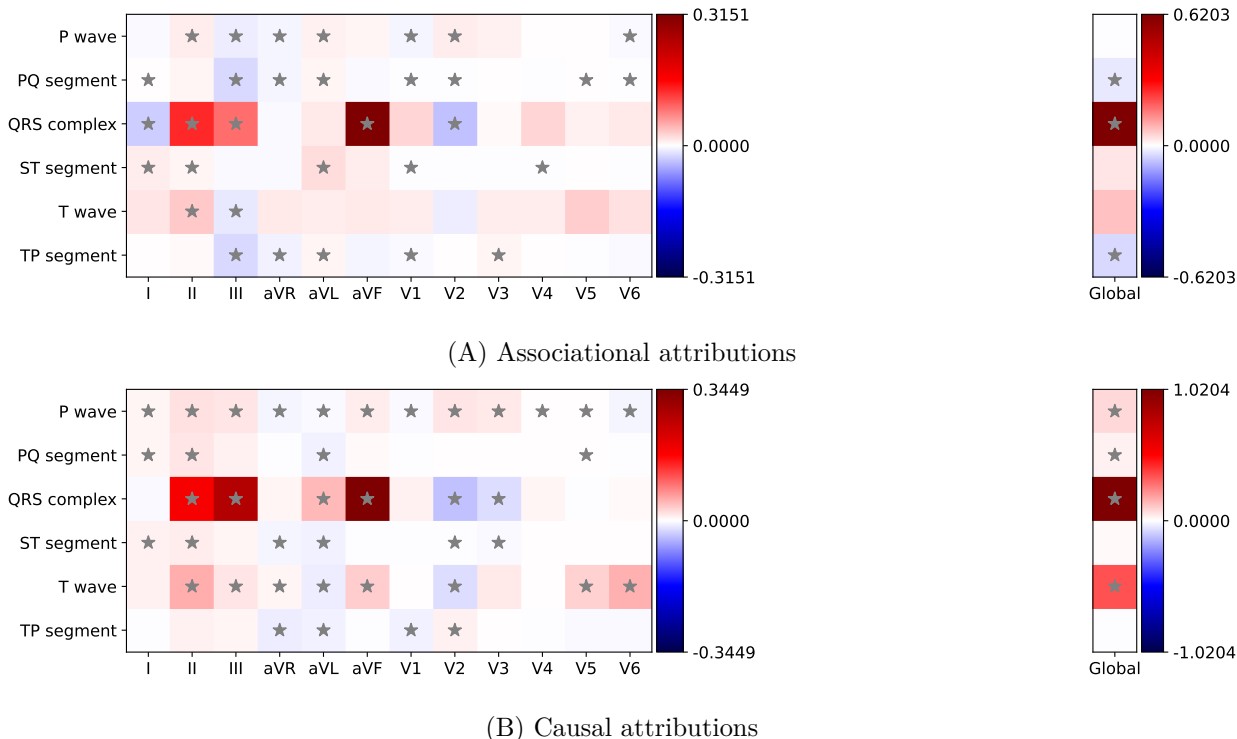

Figure 6: Illustration of the (A) associational and (B) causal attributions on the PTB-XL dataset

appropriately highlight positive effects for the QRS complex in leads II, III, and aVF, which have been linked to pathological longer and deeper Q-waves (Thygesen et al., 2018). In the associational attribution map, a negative significant effect is observed in the T-wave for lead III, while the causal attribution indicates a positive significant effect. Literature works align in this case rather with the causal attribution in the sense that high T-waves exhibit a positive pattern (Dressler & Hugo, 1947). Similarly, literature results suggest a positive effect for the P-wave in leads I, II, and III (Grossman & Delman, 1969), which are recognized as significant and positive effects from causal attributions, while associational attributions only show significant positive effects in II and a negative effect in III.

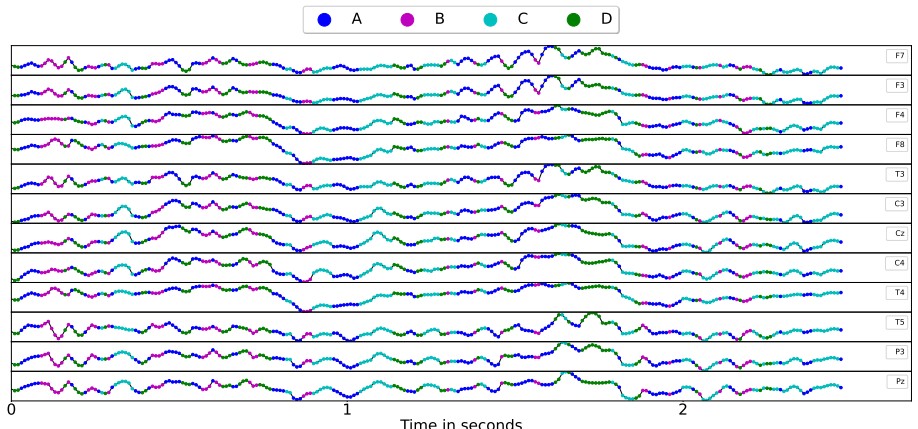

Figure 7: Schematic representation of the concepts for the schizophrenia dataset

**EEG classification** As the third dataset, we analyze the schizophrenia dataset (Borisov et al., 2005), which includes EEG signals from a study involving paranoid schizophrenia patients and healthy controls. This

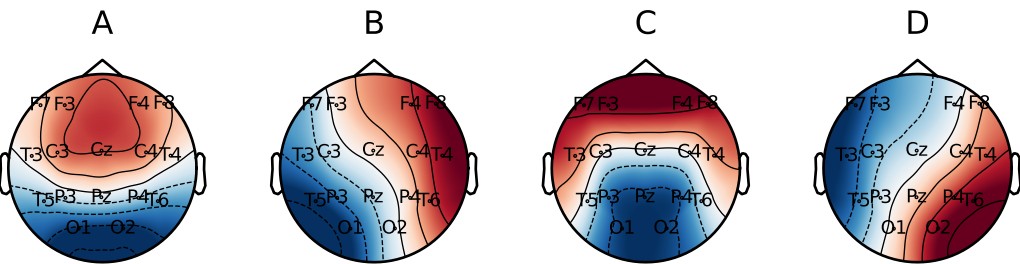

Figure 8: Spatial distribution of brain activity patterns during different states of brain processing. Dark red indicates increased activity, while dark blue signifies decreased activity.

dataset comprises 16 EEG channels (F7, F3, F4, F8, T3, C3, Cz, C4, T4, T5, P3, Pz, P4, T6, O1, O2), with each channel spanning 248 time steps. Further details on the dataset and preprocessing are available in the supplementary material. To extract meaningful concepts, we employ an EEG microstates analysis (Pascual-Marqui et al., 1995) through open-source software (von Wegner, 2017; Gramfort et al., 2013). These microstates capture transient brain states reflecting underlying neural dynamics, often linked to specific cognitive processes. Our analysis identifies four distinct concepts (A-D) leading to a concept validation score (AUROC) of 0.8249 (95% PI 0.7682-0.8793). As a supporting illustration to compare our findings with the literature, we present in Figure 8 a topographic map illustrating the overall brain activity during each investigated EEG microstate. We report an AUROC classification performance for the S4 model of 0.9671 (95% PI 0.9432-0.9849). Figure 7 shows an exemplary visualization of the concepts for a schizophrenia sample.

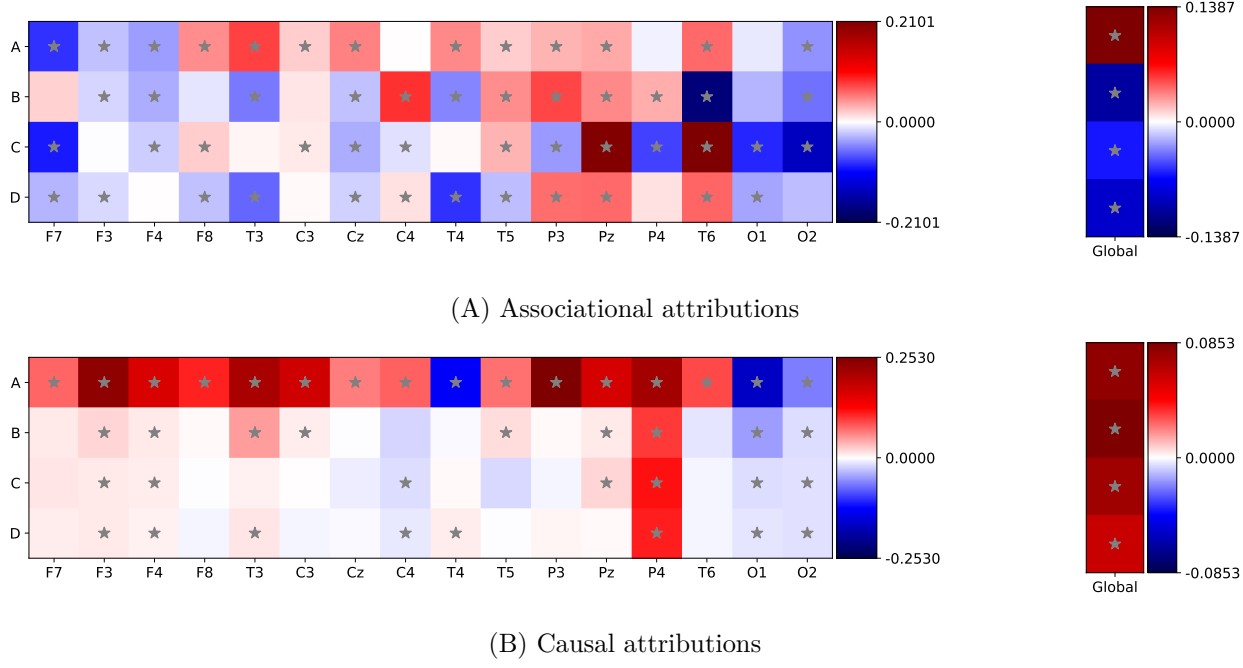

(A) Associational attributions

(B) Causal attributions

Figure 9: Illustration of the (A) associational and (B) causal attributions on the schizophrenia dataset

Figure 9 presents the associational and causal attributions for the EEG classification task. Several studies in the literature have identified specific patterns associated with schizophrenia. From a global perspective, B exhibits statistically significant differences between patients and controls in numerous studies, considering both duration (Kikuchi et al., 2007; Koenig et al., 1999; Nishida et al., 2013) and occurrence (Koenig et al., 1999; Nishida et al., 2013). Moreover, other studies have highlighted the importance of A and C based on features such as occurrence, coverage, and duration (Keihani et al., 2022), as well as D due to increased

mean duration (Sun et al., 2021). Thus, while associational attributions do not adequately cover all expert knowledge attributions globally, causal attributions do. From a channel-wise perspective to the best of our knowledge, we are the first work to investigate any effect of single leads microstates for schizophrenia detection using EEG. In the two previous datasets, the concepts typically exhibit a consistent pattern across channels, however, here the associational plot appears to show random behavior.

## 5    Discussion and conclusion

**Limitations** At this stage, *CausalConceptTS* faces several limitations, which we briefly discuss in the following. First, our method does not account for intervening on the concept mask $M$ i.e. different concept length for specific class but relies on a predefined mask from the original sample. This could pose issues, especially for pathologies, like the left bundle branch block in the ECG case, which is characterized by a wide QRS complex, i.e., altering the concept mask significantly. To mitigate this, one could consider combinations of adjacent concepts instead of individual ones. Second, the generative model for imputation is trained solely on real samples, assuming it generalizes well to unseen classes when conditioned on concepts from other classes. Third, intervening on specific concepts with a different class inevitably requires evaluating the model slightly outside its model scope, blending characteristics of the original class and the intervened state. Fourth, the proposed approach only focuses on the causal effect of the considered concepts but does not incorporate other possible confounding factors, such as static patient metadata like demographic data, see the discussion below Eq.5. Their impact could be investigated by explicitly conditioning the generative model on these variables.Finally, an extensive analysis of channel correlations, which is closely related to the question of interaction effects (Blücher et al., 2022), both from an associational as well as from a causal point of view, is beyond the scope of this work but represents a pressing direction for future research.

**Different classification scenarios** For multilabel classification, where multiple labels might be present in a single time series sample, one could adapt the framework by defining separate interventions $D_i^*$ for each label $i$ alongside the baseline intervention $D^0$. This would allow to analyze label-specific causal effects by computing $\mathrm{do}(D_i^*) - \mathrm{do}(D^0)$ for each label $i$ where the sample-level attribution could be the agregated value of the individual attributions. For multiclass classification, where each time series sample belongs to one of several mutually exclusive classes, we can generalize the intervention $D^*$ to represent each specific class $k$. The causal effect for a given class $k$ would then be $\mathrm{do}(D_k^*) - \mathrm{do}(D^0)$, see also the related discussion in (Goyal et al., 2019).

**Use-cases for XAI and broader impact statement** As described in (Wagner et al., 2024), one has to distinguish different use-cases for XAI such as providing side-information for end-users, model auditing, and knowledge discovery. While the first use-case can only be assessed with extensive user studies, the two latter use-cases rely on the fact that the used attribution method faithfully captures the model behavior. In this case basing auditing decisions or claiming discoveries on potentially misleading associational attributions represents a danger. In any case, it is worth acknowledging the difference between causal and associational attributions and taking this aspect into account when assessing the suitability of particular attribution methods in particular in safty-critical application domains.

**Conclusion** The paper proposes a framework to assess the causal effect of class-specific manifestation of predefined concepts of a time series on a given fixed time series classifier. Its key component is a high-fidelity diffusion model, which is used to infer counterfactual manifestations of concepts under consideration. This allows us to compute individual and average treatment effects. Furthermore, we demonstrate that the main difference between such causal attributions and purely associational, perturbation-based attributions lies in the use of a class-conditional as opposed to an unconditional imputation model. These insights allow for a direct comparison of causal and associational attributions. The differences between causal and associational attributions hint at the danger of drawing misleading conclusions from associational attributions. We showcase our approach for a diverse set of three time series classification tasks and find a good alignment of the identified causal effects with expert knowledge.

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
