# OpenReview forum: "CausalConceptTS: Causal Attributions for Time Series Classification using High Fidelity Diffusion Models"
_TMLR — Rejected by TMLR_

### Review · Reviewer_SBtN · 2024-09-27

**Summary Of Contributions:**

The paper proposes an approach to XAI, specifically the post-hoc explanation of the performance of a classifier constructed from time series data. The approach taken is meant to be from a causal perspective, wherein the goal is to estimate the causal effects of concepts/features on classification outcomes. The approach is agnostic about the origin/type of classifier.

Ultimately, the paper proposes an estimator for causal effects of "concepts" defined from time series (which I take it corresponds to a possibly labeled region of the observed time series) on an outcome which is the output of a classifier, presumably trained on the same or similar time series data.

**Audience:**

No

**Claims And Evidence:**

No

**Requested Changes:**

In my opinion, the paper would need to be re-written with more clarity, more standard notation, and precision. The estimation procedure should be more formally described, and its properties formally evaluated (i.e., under what conditions it performs as expected).

Since among the review criteria for TMLR, clarity and technical correctness are emphasized, the paper must be re-written to be clear enough that technical correctness can be evaluated.

**Strengths And Weaknesses:**

Strengths:
- The paper addresses an important problem: explanation of classifier models is potentially very impactful if it can be done reliable, and has implications for algorithm auditing.
- The paper takes a causal perspective, which has been proposed in some earlier work on XAI, but there are still very few works in this domain.

Weaknesses:
- I personally find the description of the proposed methodology very unclear and insufficiently rigorous. The notational choices and vocabulary are opaque and mostly non-standard (speaking from the perspective of a causal inference researcher in statistics/computer science/ML). The methodology is described in Section 3. The ingredients of the data-generating process are described, but in a way that I struggle to understand what the various objects refer to. The final observable "signal" $X$ is constructed from a "mask" $M$ ("of the same shape as the signal") and some variables labeled $X^c$ where these denote "the actual numerical values $X$ takes in segment $c$." The authors define $X \equiv X(X^1,...,X^C,M)$ though I do not recognize what this means exactly (I guess that $X$ is a function of these ingredients?). I unfortunately do not understand the set up here at all: what is the mask exactly, what does it mean for this to have "the same shape" as $X$?), what is $X^c$ and how it can be both giving rise to $X$ but also defined in terms of $X$?... The data-generating process is not clear to me. (The classifier $f$ is also not at all explained: what outcome is it classifying, and from what data is it trained?) Next, the authors define an "individual treatment effect" in a way that is very opaque and not the standard way treatment effects are defined in the causal inference literature. (This is not the definition used in the reference cited for example, which defines the ITE=CATE=$E[Y(1)-Y(0)|X=x]$.)
- The procedure for estimating the target quantity is also not clearly or rigorously defined. The authors describe a model architecture informally, but it is not clear how the pieces add up to an estimator of the target causal effect. The level of description given certainly would be insufficient for anyone to reproduce the same estimator. (Various details are only gestured at, e.g., "sampling from an appropriate generative model.") It is not clear what properties this procedure has -- is there any guarantee that the proposed estimator is consistent, i.e., converges to the true target parameter as the time series grows? The authors refer to using bootstrap to generate intervals (unclear if these are prediction intervals or confidence intervals) but the bootstrap is valid under specific conditions which are certainly violated here. What guarantees at all are there about the reliability of the procedure? No theory is presented and the properties of this procedure are unclear.
- There is no clear discussion of how confounding (which after all separates causal from associational conclusions) is addressed.
- Ultimately, for the reasons above, it is difficult to evaluate the technical correctness of the methodology presented in this work. I am not able to confidently state the conditions, if any, under which the proposed estimator approximates the target quantity or supports formal statistical inference (valid significance tests or confidence intervals).
- The description of prior work is so brief and jargon-y that it is not informative. It is hard to get a sense of what the other approaches mentioned actually do.
- There are also other, more minor weaknesses, but I will not raise all of these here, since review time is limited and I think the above are sufficiently fundamental.

---

> ### Author Response · Authors · 2024-11-01
> **Response**
>
> 1)	I personally find the description of the proposed methodology very unclear and insufficiently rigorous.
>
> 1.1 We sincerely appreciate the reviewer’s comments as the manuscript was primarily written with an XAI audience in mind. Any hints on how to make it more accessible also for the causal inference community would be very much appreciated. We have updated our draft and completely rewritten the part on the data generating processing in order to improve the readability of the manuscript. The semantic segmentation mask M serves as an intermediate (first) step during the data generating process, where we assume that M is generated first. We assume that only in a second step the numerical entries of the time series are generated in subsets that align with the unique entries in M.
>
> 1.2 We also tried to clarify the notation on how the X^c_M relate to the total signal X. The intuition is to insert the numerical values X^c_M at the appropriate places where the segmentation mask takes the value c.
>
> 1.3 Regarding the classifier question about what outcome is it classifying, and from what data is it trained, we assume a predefined (binary) classifier, which takes a time series X as input. Whereas we consider classifiers f as deep neural networks, the approach does not rely on that and would work equally for classical ML classifier operating on features extracted from X. As stressed in a new paragraph in the experiments section, the goal is to investigate the classifier f through probing with counterfactual samples.
>
> 1.4 Finally, in regards to the equation of the individual treatment effect, now is more better aligned towards the standard definition in the literature with do operators, similarly we have omitted the mask in the equation explicitly but rather include within other variables (however still discussed its presence in main text) for better readability.
>
>
> 2 and 4) (The procedure for estimating the target quantity is also not clearly or rigorously defined.) (Not able to state the conditions under which the proposed estimator approximates the target quantity or supports formal statistical inference).
>
> We thank the reviewer for the valuable comments. From our point of view, the issue is rooted in the fact that we use an empirical imputer to approximate the true (but unknown) data generating process. This process is not as uncontrolled as it might seem at first glance. As described in the revised manuscript, one can infer prediction intervals from multiple imputations and turn them into statistically valid prediction values using conformal prediction techniques (even though this is beyond the scope of this submission), at least for the in-distribution case where the class to be imputed coincides with the sample’s true class. Deriving statistically rigorous prediction intervals for the counterfactual case is unavoidably much more challenging. The reason is that a class-conditional imputer model will be conditioned on input from a different class.  In our case, we rely on the ability of the imputer model to generalize slightly out of its training domain. As there are no formal guarantees in this case, we at least report statistical uncertainties across samples using bootstrapping to demonstrate the robustness of our findings.
>
> 3) No clear discussion on how confounding is addressed
>
> We thank the reviewer for highlighting this aspect. In our work, we focus on isolating causal effects to improve upon traditional associational methodologies by directly addressing potential confounding factors. Specifically, our methodology integrates causal inference tools, such as do-calculus and conditional independence assumptions (based on Pearl’s framework, 2009), which allow us to adjust for observed confounders and better approximate the true causal effect. By conditioning on the intervention $D = D^0$, we effectively control for confounding pathways coming from the class state at specific concept-mask pairs that might otherwise bias the results (see updated equations in manuscript which showcase a clear comparison of associational vs causal). Any remaining confounders, such as unmeasured/unobserved static patient metadata, are acknowledged as limitations at this stage in our methodology, yet we further discuss their implications in the manuscript, and it is certainly are a significant future work to research.
>
> 5) the description of prior work is so brief and not informative. Hard to get what the other approaches mentioned actually do.
>
> We thank the reviewer for the observation. We tried to improve the readability of the related work section and added short paragraphs to put literature works into the perspective of the proposed approach.

---

> ### Comment · Reviewer_SBtN · 2024-11-16
> **Comments after revision**
>
> I thank the authors for engaging with my comments and updating their manuscript to try and clarify the issues raised. I recognize this takes a lot of work and commend the authors for this.
>
> However, unfortunately, in my opinion the revisions do not sufficiently address all my concerns with this paper.
>
> 1. Though the revised paragraph at the bottom of pg 3 clarifies some points, I still find it formally quite difficult to understand the details here. I better understand what CS and M refer to now, though I'm still confused by what X = X(X^1_M,...,X^C_M) means (that X is a deterministic function of these elements?) and what X^c_M[k] means since X^c_M is a *set* of numbers according to the definition above... in other words, I struggle to make formal mathematical sense of the model here. (The formalism is also quite different from Scholkopf et al 2012 so I'm not sure why that is cited. If the presentation mirrored more closely Goyal et al 2019, a paper which does use "concepts" as the relevant causal variables, it would be clearer to me.) In any case, it seems like some of the other reviewers are more comfortable with the presentation here so I won't dwell on this anymore.
>
> 2. The definition of the target causal effect quantity in (1) is still not right to me, even with the use of the "do" notation. It seems that the authors have in mind an intervention on CS that holds the value of M "unchanged" -- this would suggest that the target quantity is actually a mediation effect (the natural direct effect, NDE) not an ITE.* In the standard causal formalism, if we have an exposure A (here CS) and a mediator M (here the mask) and an outcome Y (here f(X)), the natural direct effect of A on Y where M takes on its "natural" value would be written: E[Y(a1,M(a0))] - E[Y(a0,M(a0))]. (Values a0, a1 stand in for values CS*, CS0 here.) This might sound pedantic or like a simple notational preference, but in my opinion it is important to be clear about what the relevant quantity really means and also whether it is identified. I'm not against using the "do" notation at all, but when it comes to these kinds of mediation quantities, the standard "do" notation is known not to be sufficiently expressive -- it cannot express the fact that M is held at the value it would take "in another world" -- which is why for example Pearl and other authors in causal inference use a version of the potential outcome notation above when defining direct and indirect effects. It is also important to note that the natural direct effect is not generally identified in the presence of mediator-outcome confounding, so in the model specified in Figure 2, this quantity is not identified due to the confounder between M and its descendent.
>
> *Note: an ITE, when used as synonymously with the CATE, is written: E[Y(a1)-Y(a0) | Z=z] where Z are baseline covariates -- this quantity is a *function* of baseline covariates. But the equation (1) is not. The authors cite Shalit et al 2017 but even the definition (4) in that paper is equivalent to E[Y(a1)-Y(a0) | Z=z], not what the present authors write as (1).
>
> If the target of intervention is CS and CS has no causal parents (Figure 2), then there is no (measured or unmeasured) confounding for the average causal effect; the average causal effect simply equals the conditional probability of the outcome given CS, P(Y | do(CS=CS*)) = P(Y | CS=CS*). Then there is no real "causal inference" to do, because under that set of assumptions, the causal effect just equals an association. But I think the authors actually have in mind the NDE, not ITE.
>
> Ultimately, there seems to me to be a fundamental confusion or lack of clarity about what is the quantity being estimated. And moreover, if the quantity is indeed the natural direct effect, there is an expansive literature on estimators for the NDE.
>
> 3. The authors' response to my point about estimation guarantees (consistency of the procedure, i.e., that it converges to the "true effect") and quantifying uncertainty does not alleviate my concerns. Prediction intervals are not relevant here, since the target is a causal effect -- for that one would want a confidence interval (or Bayesian credible interval) for the target quantity, something which shrinks to the true value as sample size grows to infinity. Standard bootstrap is not guaranteed to be valid in this setting, so it is not clear what properties the stated "interval" has. Also, as already pointed out the steps of the actual procedure are not described in detail, so it would not be straightforward to reproduce the estimator or study its properties. In my view, it is important to have some sense that the procedure, in this case the output of a complicated generative model, actually converges to the truth (or under what conditions this is so).

---

### Review · Reviewer_6WKN · 2024-10-07

**Summary Of Contributions:**

The submission proposes a model-agnostic framework to estimate the causal effects for predefined segments in time series across multiple channels. Specifically, the authors adopt a slightly different definition of individual/average treatment effect from conventional causal inference literature by calculating the logarithmic difference of the expected outcome over data-generating process $h_X^c$ for control and intervened samples, where the distribution of $h_X^c$ is modeled by state-space diffusion. The authors compare associational and causal attributions estimated by the proposed framework on 3 meteorological and physiological datasets and show that the estimated causal attributions align better with the findings in the existing literature.

**Audience:**

Yes

**Broader Impact Concerns:**

The authors have appropriately included a broader impact statement in this paper.

**Claims And Evidence:**

Yes

**Requested Changes:**

The authors may wish to improve the overall flow of the manuscript by providing additional clarifications and rationales in the presentation of the methodology. Please see the weaknesses for details. Moreover, while the experimental results appear reasonable, it would be beneficial to include a synthetic experiment. This would help demonstrate that the proposed framework can accurately estimate the **magnitude** of the ITE/ATE, rather than merely predicting the directionality (positiveness or negativeness).

**Strengths And Weaknesses:**

**Strengths**:

•	Causal attribution and reasoning in time series data is an important area of research, particularly in healthcare and medical domains, and warrants further exploration.

•	The authors have made the implementation of their method publicly available, which enhances the accessibility and reproducibility of their work.

•	The visualizations of associational and causal attributions presented in the experimental results offer an effective comparative approach. Furthermore, the authors have substantiated the validity of their estimated causal effects by aligning their findings with established results in the existing literature.

**Weaknesses**:

•	The authors do not provide a clear explanation of the actual meaning of the noise variables $\epsilon_D$ and $\epsilon_S$, which are used in the generation of the disease state $D$. Additionally, the rationale for incorporating these two variables is not addressed, and they are not visualized in Figure 2 as well.

•	Following the point mentioned above, how shall we intervene on $D$ and set it to a specific value value $D^*$ if $D = g(\epsilon_D, \epsilon_S)$?

•	The definition of ITE in Eq. 1 is not accurate enough. Specifically, both $X_c$ and $M$ are dependent on the disease state $D$, but Eq. 1 only reflects the dependence with respect to $X_c$.

•	It seems that the terms “concept”, “class”, and “segment” are used interchangeably, but they all refer to the same thing (i.e., predefined segments in time series), which can probably lead to confusion.

•	While the authors argue that the requirement for predefined concepts should not be viewed as a constraint of the proposed approach, I respectfully disagree. Even if concepts are identified using clustering algorithms, there remains a lack of interpretability regarding what each concept represents and how their relative importance is determined. This poses significant limitations on the method’s applicability in real-world scenarios.

•	From Figure 3, it seems that the number of clusters for K-means is not optimally chosen, as the time points corresponding to concept B are much fewer compared to those corresponding to other concepts.

Minor comments:

•	There is a small typo in Figure 1. The term under $ITE_i$ should be $P(X_i | do = 0)$ instead of $P(X_i | do = 1)$.

•	There is a right parenthesis missing on the second last line on page 3.

•	There are two consecutive commas in Eq. 3.

•	The symbols $X_c$ and $X^c$ are used interchangeably.

---

> ### Author Response · Authors · 2024-11-01
> **Response**
>
> 1) The authors do not provide a clear explanation of the noise variables.
>
> We thank the reviewer for the observation. Our incentive was to demonstrate the full data generative process including the way the disease state (called class state in the revised manuscript) was created. However, based on the reviewers’ comments, we realized that this caused a lot of confusion and decided to remove it from the description (as it is also not relevant for the intervention part). The remaining noise variables \epsilon^c_X can be understood as an inherent source of randomness in the process of generating signal fraction X^c_M for a given mask M and class state CS, which is later captured through the probabilistic nature of the generative imputation model.
>
> 2) Following the point mentioned above, how shall we intervene on D...
>
> We thank the reviewer for the comment. We believe it strongly aligns with the answer to the first issue. In the updated manuscript, the data generating process starts from a disease state (now called class state CS) and we intervene precisely on the level of this class state. We apologize for the confusing wording and update the draft appropriately for better readability.
>
> 3) The definition of ITE in eq. 1 is not accurate enough
>
> We appreciate the reviewer’s observation. We rewrote the corresponding sections to improve the readability.
>
> 4) It seems that the terms “concept”, “class”, and “segment” are used interchangeably....
>
> We thank the reviewer for this observation. We have renamed several terms to improve the readability of out manuscript. The term ‘segment’ have been updated to the term ‘concept’ in all the cases including ‘concept mask’ instead of ‘segmented mask’ except for the definition of concepts (‘i.e. segmented sub-series’). The only other exception is the ECG example, where the concepts are conventionally referred to as ECG segments.
> While the terms concept and segment can be used interchangeably, it is important to differentiate them from the term “class” (state). Every sample comes with a predefined underlying class state. We later intervene on this class state to derive counterfactual outcomes.
>
> 5) While the authors argue that the requirement for predefined concepts should not be viewed as constraint...
>
> Thank you for your valuable feedback and we fully share the reviewer’s concerns about the interpretability of concepts derived via clustering and extended the corresponding paragraph in the manuscript. We believe that a second concept validation step is necessary to assess the semantic meaning of the identified clusters/concepts. From our understanding, the way this second step will be carried out will depend very much on the particular domain of interest. We would also like to stress that for example in the EEG domain, microstates/concepts are conventionally inferred via clustering and then assessed based on their localization properties. In fact, we present such an analysis in the EEG experiments in our manuscript demonstrating the feasibility of this step in certain domains.
>
> 6) From figure 3, it seems that the number of clusters for K-means
>
> We thank the reviewer for your comment regarding the number of clusters in K-means as shown in Figure 3, where we inferred the most suitable number of clusters via the elbow method. We appreciate your observation about the uneven distribution of time points across the concepts. The K-means algorithm tends to group data based on the inherent structure and density of the data points rather than aiming for an even distribution. In our case, concept B represents a distinct set of features that naturally occurs less frequently in the dataset, leading to fewer time points being allocated to that cluster. This uneven distribution reflects the actual characteristics of the underlying data rather than a limitation of the clustering method itself. Visually and numerically, we believe that this clustering effectively highlights the differences among the concepts, even if the number of time points is not equal.
>
> 7) Minor comments: typo in figure 1, right parenthesis missing, two consecutive commas, and interchangeable use of X_c and X^c.
>
> We thank the reviewer for the observations. We have corrected these minor issues which improved the readability and correctness of the manuscript.

---

> ### Author Response · Authors · 2024-11-01
> **Response continuation**
>
> 8) Suggestion: synthetic dataset to estimate the magnitude of the effects
>
> We thank the reviewer for the suggestion. We fully agree that additional experiments based on synthetic datasets would be really instructive. As a first step, we provided an explicit toy example for a single concept, which already demonstrates a crucial difference between causal and associational attributions.
> From our perspective, a central challenge seems to be define a proper ground truth for the attribution. While at least the sign of the attribution is often uncontroversial, the magnitude of the attribution is typically hard to assess. We believe the extension of the toy example towards a more realistic scenario would have to encompass several directions, such as correlations between different concepts within the data generating process and imperfect classifiers that at least partially exploit such correlations. We found it challenging to define such a synthetic example that incorporates these aspects while still being representative for real-world classifiers. We hope that at this stage the toy example helps to clarify the inner pitfall of associational attributions. However, we would very much appreciate additional ideas in this direction.

---

> > ### Comment · Reviewer_6WKN · 2024-11-15
> >
> > Thank you for addressing my questions and thoroughly revising the manuscript. After reviewing the rebuttal and the corresponding revisions, I find that the readability of the paper has improved, and the authors have largely addressed my concerns. I also briefly reviewed the comments from the other reviewers, and it seems that a major point of concern is the clarity of the methodology, particularly whether the terminologies are clearly defined. In my opinion, the authors have made substantial efforts to enhance the clarity of the paper, making it more accessible to readers with limited background in causal inference in time series. However, I will take the opinions of the other reviewers into account before finalizing my recommendation, should there be additional feedback.

---

### Review · Reviewer_EyBC · 2024-10-24

**Summary Of Contributions:**

The paper presents a novel approach to explainability in time series models, focusing on concept-based explanations. Time series explainability is an important application with very few prior work that investigate the problem with rigour in particular with a causal perspective. This work introduces "concepts" in a time series setting, which to my knowledge is a novel idea, and explains the model's decision making through these concepts. This work also has an interesting analysis and comparison of causal versus associational explanation.

**Audience:**

Yes

**Broader Impact Concerns:**

This work has no major ethical concerns to the best of my knowledge. However, I strongly believe that claims related to causality, in particular in healthcare applications, should be rigorously evaluated.

**Claims And Evidence:**

No

**Requested Changes:**

1. This paper requires further refinement and proofreading. There are several typographical errors that should be addressed. For example, the last sentence of the first paragraph in the "Related Work" section and the last sentence of the third paragraph contain typos. Additionally, in Figure 1, the mean output probability for the baselines and the do operation has a typo. Overall, I think the paper can be improved in terms of readability.

2. Some of the points I mentioned in the earlier sections are critical to show the utility of the work and prove the claims in the paper. Especially, a more comprehensive evaluation (point 4), and more discussion on counterfactual samples (point 2).

**Strengths And Weaknesses:**

While the proposed method is innovative, several areas require further clarification and development to improve its practical utility. In this review, I will highlight key issues, if addressed, could significantly enhance the robustness and applicability of the proposed framework.

1. Definition of "Concepts": The proposed method relies heavily on the notion of "concepts" for explaining the model outcome. While the authors state that the discovery and evaluation of these concepts are beyond the scope of the paper, I believe this is a critical aspect that affects the utility of the framework. The framework's explanations will inevitably be influenced by how concepts are defined. In many settings obtaining labels for concepts can be difficult or unrealistic. In addition, the concepts should be defined in a way that are meaningful for explaining a particular model. Now in cases where we don't have expert annotated concepts, the proposed solution is to use K-means for self-supervised discovery of concepts, which raises further concerns. Specifically, how are the number of clusters determined, and how can we ensure that these clusters represent meaningful concepts for explaining the model? It would be a major strength if the explainer model itself could automatically determine relevant concepts, rather than relying on predefined labels or clusters.

2. Counterfactual Samples: The generation and evaluation of counterfactual samples are known to be challenging tasks, and I commend the authors for employing a diffusion architecture, which is well-suited for such tasks. However, given that the method relies heavily on the generator's performance, it would be beneficial to include an analysis of the model's sensitivity to the generator's quality. A more significant concern is the assumption that only one concept is masked in the counterfactual samples, meaning the intervention is expected to change only that concept. In real-world time-series data, interventions often affect multiple aspects of the series simultaneously. As a result, the counterfactual samples produced by the proposed framework may not be realistic and could fall outside the distribution for which the classifier was trained, reducing the reliability of the explanations. Additionally, the assumption that the masks are the same for every intervention is not always valid. For example, in ECG data, an intervention might affect the entire QRS complex, changing the location (mask) for different concepts.

3. Use of the Term "Causality": The term "causality" should be used more carefully in the context of explainability. The framework is explaining the model’s decision-making process, but the causal relationships derived are based on the model’s learned assumptions, not necessarily the true causal relationships within the data. It is essential to distinguish between model-driven causality and the underlying data's causal structures, especially in sensitive fields like healthcare.

4. Evaluation and Comparison to Attribution-Based Methods: The evaluation section would benefit from a more thorough comparison with attribution-based explainers, at least including one baseline for reference. The current evaluation is lacking comprehensive analysis, and the claims that the causal explanations are superior due to the existence of known relationships require stronger empirical backing. A direct comparison with established attribution-based methods would make the results more compelling and grounded.

5. Additional Discussion: A few points could enhance the discussion section. First, the framework could be extended to non-binary settings, which would broaden its applicability. Second, patient static data, such as demographic variables, have a significant impact on time-series data. Since these features are often available, conditioning the generator directly on these variables—rather than treating them as unobserved noise—could lead to more accurate and meaningful counterfactual samples.

---

> ### Author Response · Authors · 2024-11-01
> **Response**
>
> 1)	Definition of concepts
>
> We thank the reviewer for the observations. In regards concepts discovery, the number of clusters was determined using the elbow-method as stated in section number 3 under ‘concept discovery and concept validation’. Here, we prioritized flexibility given that most of the time series data lack of well-defined concepts. Compared to alternatives like frequency-based analysis our approach a distinct advantage in adaptability, as it remains versatile across datasets without requiring prior assumptions about periodicity or stationarity, furthermore, we decided on k-means among different clustering methods based on a simple yet effective benchmark on microstates explained variance for EEG data.
> With regard to the question how one can ensure these represent meaningful concepts for explaining the model, our concept validation approach allow us to verify if even simple statistical profiles can reliable distinguish between classes, which certainly even a simple set of these showcase good discriminative performance.
>
> Finally, concerning the an automatically determination of relevant concepts rather than relying on predefined labels, we believe in modular frameworks where a (causal) concept attribution framework can be combined with any available concept discovery and validation framework. We agree that developing new methods for time series concept discovery could be a valuable future research direction, extending beyond interpretability alone.
>
>
> 2)	Counterfactual samples
>
> We thank the reviewer for the observations.
>
> First, with respect to the sensitivity of the generator, we recognize the dependency of counterfactual fidelity on the robustness of the underlying generative model. Therefore, we have used a state-of-the-art diffusion model for time series modeling. At least for the in-distribution scenario, where the class label of the segment to be imputed coincides with the sample’s class label, it is known from prior work that these models are quite robust, see the detailed analysis see (Alcaraz and Strodthoff 2023a) as cited in the manuscript. We fully acknowledge that the ability of the generative model to generate counterfactual samples unavoidably relies on the ability of the generative model to generalize outside its training domain. For example, a class-conditional generative model will be conditioned on samples from a different class, see also the reply to the second issue raised by reviewer SBtN.
>
> Second, we appreciate the reviewer’s comments concerning the fact that we only consider single concept relevances. The framework would indeed allow to also assess the joint attribution of multiple concept. This would in fact be a very interesting direction for future work also with regard to the analysis of interaction effects, see (Blücher et al 2022) for a treatment for associational attributions.
> Third, we fully agree on the shortcomings of the assumption that the underlying segmentation mask is not adjusted based on the class state. A specific example mentioned in the discussion section is that of a left bundle branch block in the ECG domain, which is characterized by a broadening of the QRS complex. To a certain degree this limitation can be mitigated by considering several adjacent concepts in the segmentation mask. However, to deal with this property requires an extension of the proposed approach. We believe that the proposed approach provides a sensible first approach for extensions beyond widely considered associational attributions and therefore decided to clearly stress this shortcoming but not to extend the framework to address it in this work.
>
> Finally, with respect to the same masks for every intervention we agree with the reviewer, the masking location and length indeed vary for each sample, therefore, we have updated the text to clarify this.
>
>
> 3)	Use of term “causality” in explainability.
>
> We thank the reviewer for the observation. We agree on the distinction of the definition of both outcomes. We have updated the draft with a corresponding disclaimer at the beginning of the experimental section.

---

> ### Author Response · Authors · 2024-11-01
> **Response continuation**
>
> 4)	Evaluation and comparison to attribution-based methods
>
> We thank the reviewer for the comment. We are fully aware of the broad range of attribution methods, which sometimes even lead to conflicting insights. The idea behind our work was to provide a first direct assessment of associational vs. causal attributions. Therefore, we used the associational attribution that is a direct analogue of the proposed causal attribution in order to enable a like-by-like comparison. We envision that it might be possible to also devise causal variants of other associational attribution methods such as Shapley values, but consider this topic clearly beyond the scope of our initial assessment. Therefore, we decided to only acknowledge the challenges arising from the large number of possibly disagreeing attribution methods in a disclaimer paragraph at the beginning of the experiments section as we are not convinced that we are already at the point where we can extend the formalism to further attribution methods.
>
> 5)	Additional discussion
>
> We thank the reviewer for the observations. We fully agree that also the extension towards multi-class scenarios would be interesting and we inserted a corresponding remark in the manuscript. However, we would like to stress that these can be accommodated within the same framework, see also (Goyal et al 2019).
> Concerning the inclusion of static metadata, we agree that these have a significant impact on time series data, nevertheless, we see this as a future research direction where we should compare in a more fine grained approach causal attributions with literature review (e.g. P-wave for males vs females), therefore, in this work, we aim to firstly demonstrate robustness towards the overall class.
>
> 6) Typos and minor issues
>
> We thank the reviewer for the observation. We have corrected the typos and minor issues.

---

> > ### Comment · Reviewer_EyBC · 2024-11-18
> > **Thanks for the response**
> >
> > Thank you for your response. The revised manuscript has improved clarity significantly.
> > The rebuttal response answers some of questions, but not all the concerns. I still believe there are elements in the proposed method  (such as the definition of concept and the limitations with masks) that limit the usability of the framework. And also, the method requires more rigorous validation. I understand and completely agree with the authors response that counterfactual sample generation is a difficult task, however, it is important to know how sensitive the explanations are to the quality of these samples (and hence the generative model used).

---

### Comment · Action_Editor_GsUW · 2024-10-24
**Authors, please consult the reviews**

Dear Authors,

Thank you for your patience as we recruited reviewers with appropriate expertise to evaluate your work. We have now received three reviews for your paper. The reviewers have some split opinions that may be amended according to your responses and updates to the submitted paper.

Please spend some time responding to the comments and/or correcting any misunderstandings that may have arisen. The discussion period is now open and the reviewers will be asked to submit a formal recommendation for this paper in two week's time.

Best,
Action Editor

---

> ### Author Response · Authors · 2024-11-01
> **Response**
>
> We sincerely appreciate the reviewers' positive feedback and the support of the action editor, comments, and constructive criticisms, which have significantly improved the quality of the manuscript. Below, we address each of the issues raised in detail. For the reviewers' convenience, we have included a marked-up version of the updated manuscript, with all changes clearly highlighted. We hope the manuscript, in its current form, is now suitable for publication in Transactions on Machine Learning Research (TMLR).
>
> We fully recognize the distinction between the two main areas our manuscript intersects—causal inference and explainable artificial intelligence. Bridging these fields has indeed been challenging, particularly in adapting terminology from both domains. However, after thoroughly revising and updating the manuscript, we believe its readability has improved, making it accessible to a broader audience from a general perspective. Thank you for your continued guidance throughout this process.

---

### Comment · Action_Editor_GsUW · 2024-11-08
**For Reviewers**

Dear Reviewers, please consult and respond to the author comments and revisions made to their draft. The authors have attempted to address your concerns and account for your requested changes.

I will not formally consider any recommendation you make on this paper until you have signified with a comment to the authors that you have read their responses to your questions.

---

### Decision · Action_Editor_GsUW · 2024-12-10

**Recommendation:** Reject

**Comment:**

The consensus among reviewers was that there are a sufficient number of elements used to describe and introduce the claims made by the authors when presenting the proposed framework that require additional clarification. Each reviewer commented how the revision provided by the authors was a significant improvement and felt that the work needs another round of substantial revisions (both in writing, and in formulation + experimental justification) to clarify the specific quantities that the causal framing estimates and how they are intended to be used in a generic fashion. The recommendation of the reviewers is that the paper would need to be substantially revised before consideration again in the future.

**Audience:**

The submission provides a model-agnostic framework for estimating of "concepts" in time series data. This framing would be of interest in the ML Research community and is a unique perspective on causal modeling within time series. However, there are some remaining concerns about the clarity of the assumptions and the underlying framework. In particular, the reviewers pointed out that some of the assumptions about how the individual "concepts" are defined would limit the general interest in the specific formulation presented with this work.

**Claims And Evidence:**

Following the reviewers recommendation, the proposed model-agnostic framework needs to be evaluated more rigorously, perhaps aided by a more general definition of the concepts found within the time series, to demonstrate the overall performance and utility as the paper claims.

**Resubmission Of Major Revision:**

The authors may consider submitting a major revision at a later time.